# Reframing the Refugee: Jenny Erpenbeck's Compassionate Politics

Kristian Shaw

School of Humanities and Heritage, College of Arts, University of Lincoln, Lincoln LN6 7TS, UK;
kshaw@lincoln.ac.uk

**Abstract:** Countless polls, studies and surveys conducted prior to and following the 2016 UK Referendum on Membership of the European Union confirmed immigration to be the key emotive issue for not only the British electorate, but several Western European nations. By critiquing key pieces of EU legislation, *Go, Went, Gone* (2015) by Jenny Erpenbeck offers a humanising, caustic warning of the troubling politicisation of EU and non-EU migration in Germany, suggesting the ways by which literature can destabilise institutional optics of power and counteract myths surrounding the process of racial othering.

**Keywords:** European Union; refugee crisis; Brexit; Erpenbeck; German literature

> 'Everyone is a migrant—even people who are in the same place, because that place changes over decades'. (Hamid qtd. in Milo (2017))

## 1. Introduction

In 2015, immigration was listed by German citizens as the key issue affecting the country. The salience and emotive resonance of immigration became more pressing as the humanitarian crisis of 2015—which saw a significant increase in asylum seekers entering Europe from Syria, as well as other Middle Eastern and African nations—continued to pose geopolitical problems for the European Union. Cetta Mainwaring's study of the EU's response to these developments attests that 'national borders remain a powerful, symbolic site' where migration and citizenship are concerned, with the crisis exacerbating an entrenched Fortress Europe mentality (Mainwaring 2019, p. 4). This tactic of externalising the border is evident in the Schengen Agreement (1985) and Dublin Convention (1990)—later modified into the Dublin II and Dublin III Regulations, as well as the recent introduction of the Asylum and Migration Management Regulation (AMMR)—which indicate how EU immigration and asylum policies are also defined and shaped by the politics of national security.

According to the Dublin Regulation, an asylum seeker can only have their application considered in the first EU country the individual enters, placing a huge strain on those member states with borders on the Mediterranean. While many EU member states closed their borders, German Chancellor Angela Merkel challenged this directive by advocating for an open-door policy of *Willkommenskultur* (welcoming culture) as a remedy to the political paralysis gripping the EU, a term that reshaped the German political landscape by promoting a positive attitude of acceptance towards refugees. Merkel's liberal policy, of course, not only 'reverberated across the Union but also sent shockwaves through the German political system', sparking a hostile reaction from centre-right and far-right parties (as well as anti-Islamic protests in Dresden), and leading to a surge in popularity for the Alternative fuer Deutschland party (AfD), who were already benefiting from public opposition to the humanitarian crisis (Geddes and Scholten 2016, p. 85). The AfD cultivated a negative perception of refugees in the public consciousness, fuelling further fear and animosity. As a result of this febrile attitudinal climate, Germany's asylum policies—and

immigration legislation more broadly—have consequently become far more reactive and restrictive in recent years, placing a strain on national integration.

Jenny Erpenbeck's novel, *Go*, *Went*, *Gone*, first published in 2015 and translated in 2017, anticipates that underlying tension between Merkel's *Willkommenskultur* and the rise of right-wing xenophobic populism saturating political discourse, marking a significant contribution to decolonisation and asylum discourse in European literature. The ethical politics of the novel are clear, with the narrative events inspired by Erpenbeck's own interviews with refugees following a series of protests at Oranienplatz in Berlin from 2012 to April 2014, before the introduction of Merkel's *Willkommenskultur*.[1] As Olivia Landry documents in her study of the 'OPlatz' pro-immigration movement, the transnational make-up of the asylum applicants hailing from 'Sudan, Uganda, Syria, Eritrea, Somalia, Afghanisation and other nations converted [the] square into an urban campsite that served as a loaded space for social and political discussion and negotiation' (Landry 2015, p. 399). With the eventual clearance of the camp in 2014, greater public awareness of—and outcry towards—German asylum policy led to the lifting of the *Residenzpflicht*, a law restricting the movement of asylum seekers, and improvements in their right to work. While British literature has recently witnessed a strong response to immigration and asylum policy, not least in the emergence of the Brexlit genre following the UK's decision to leave the European Union (Shaw 2021), a growing number of German novels have begun to speak specifically to multilateral forms of European asylum policy, including Joachim Lottmann's *Alles Lüge* (2017) and Abbas Khider's *Ohrfeige* (2008). In this article, I will discuss how Erpenbeck's novel goes beyond these works to reflect what I term her compassionate politics, encapsulating qualities of the *transglossic* (Shaw and Upstone 2021).

In a lecture at Cornell University in 2018, postcolonial scholar Homi Bhabha praised *Gone*, *Went*, *Gone* as a 'migration masterwork', pointing to its exposure of the barbarism of the humanitarian crisis, the legal black holes in EU asylum policy, and the politics of dishonour which continue to underpin migration discourse across periods (Bhabha 2018a). For Bhabha, the novel not only draws attention to the victims of the Dublin Agreement, but also exposes the 'jurisdictional history' underpinning the humanitarian crisis (Bhabha 2018b, p. 9). After all, *Go*, *Went*, *Gone* speaks to very particular political moments—the Second World War, the Cold War, the fall of the Berlin Wall, the legacy of German colonialism, and the emergence of the so-called refugee crisis—in accentuating the simultaneities that exist between these periods with respect to the forces of displacement, nationalism, and xenophobia. Erpenbeck's previous works, *Visitation* (2008) and *The End of Days* (2012), delve into the layers of German history to comment on the contemporary moment, but *Go Went*, *Gone* carries an urgency that highlights the vital role of the writer as a public intellectual with social responsibility. As James Wood notes in his review of the novel, Erpenbeck confronts the reader with the pressing question of 'how do we live—what should we do—once we have modified, however feebly, our colossal ignorance' of forced migration patterns (Wood 2017). Various drivers, determinants, and infrastructures underpinning migration and asylum discussed in the narrative—from neo-imperial economic sanctions that create cultures of dependency, unrelenting civil war, and unequal trade agreements reinforced by stringent EU policies towards the Global South—indicate the ways in which Western responses to migration and asylum are defined and shaped by specific cultural histories and national legacies of loss, with particular attention paid to Germany's historical culpability in contributing to contemporary migration crises.

## 2. The Challenges of Eurocentrism

From the outset, Erpenbeck's novel is marked by interiority. A third-person omniscient narrator documents the life of Richard, a recently retired Professor Emeritus of Classical Languages and member of the *Bildungselite*, as he meticulously notes the minute details of his everyday habits:

> 'Every morning, he reads the newspaper over breakfast as always. Every morning he drinks tea—Earl Grey with milk and sugar—and eats one piece of bread with

honey and one with cheese [. . .] He can take his time every day now, but he still only wants an egg on Sundays. The way he's used to it'. (Erpenbeck 2017, pp. 22–23)

Such domestic routine, defined by its repetition and triviality, marks an early tension between the privileged, predictable lives of settled inhabitants and the uncertainty that surrounds the experiences of forced migration. While watching the news on television one evening, Richard sees the report on a refugee hunger strike at Berlin's Alexanderplatz, which disrupts his self-imposed isolation. Spotting a handmade sign bearing the words 'We become visible', he realises the refugees are occupying the very spot he passed earlier that afternoon. Alarmed by his own lack of awareness of events unfolding around him, he wonders, 'Why didn't he see the demonstration?' (2017, p. 18), a question that continues to reverberate throughout the narrative.

Richard only arrives at Oranienplatz—a separate protest camp in Berlin Kreuzberg—as the camp itself is being torn down, galvanised into action following reports of hunger strikes and further boat disasters off the coast of Lampedusa; his lethargic and tentative engagement define the early sections of the novel (Richard's narration begins with the word 'perhaps'). The swift dismantlement of the protest camp—in what is a public square—exemplifies how tightly the German state protects what may be permitted to remain in institutional memory. The cultural imaginary of German political life appears seemingly unwilling to include refugee experiences into the national story (a narrative that some citizens in recent history have co-opted into a trenchant desire for ethnic homogeneity). The lack of violence or disruption at the protest is itself problematic. As one reporter explains to Richard, 'If nothing special happens, I can't make a story out of it' (2017, pp. 12–13). Violence or sensationalised events can be useful in shaping a negative narrative account of asylum, but the mere presence of the refugees highlights the structural violence being committed against them. The refusal of the refugees to give their real names indicates an awareness of the ways by which European asylum policy exploits places of origin in order to deny rights to remain, 'you have to say, they're told, otherwise how do we know whether the law applies to you and you're allowed to stay here and work? [. . .] If you were in our shoes [. . .] would you take in a guest you don't know?' (2017, p. 11). A policeman later informs Richard that the refugees will be relocated to a disused nursing home on the outskirts of town, far from the view of the local media, ensuring their presence is defined by its absence.

The precarity of the refugees reflects what Zygmunt Bauman describes as a global sense of displacement suffered by those 'left out of our sight', but due to mounting geopolitical pressures forcing European citizens to confront 'the previously comfortingly, consolingly invisible aspects of the state-of-the-world reality' (Bauman 2016, pp. 90, 91). The opening of the narrative creates a clear duality between these attempts to conceal the refugee protests from public view and the death of an unidentified man as the result of 'a swimming accident' in the lake behind Richard's house. This tenebrous death initially serves as an obvious and overt parallel for the nameless refugees drowning in the Mediterranean but comes to operate as a deeper reflection of what rises to the surface in German culture: 'As long as the body of the dead man hasn't been recovered, the lake belongs to him' (2017, p. 10). The locals' purposeful avoidance of the site, 'afraid the man would pull them down with him', is reflective of the more general cultural elision of migration and media indifference to the drowning of refugees, preventing the innumerable deaths at sea from being dredged up and confronted in public discourse (2017, p. 6). Further, this communal fascination with the mysterious death of one white male conveys how European lives are worth more than those of African refugees when viewed through the lens of racial capitalism and its subsequent production of cultural difference.

Initially, Richard remains in a state of curious voyeurism; he abruptly leaves a local debate on refugees at a school auditorium, which he attends 'for reasons unclear even to himself', when a debate on asylum becomes heated (2017, p. 28). Richard's stance is not unexpected in light of the prevalent public mood, but he is well placed to question

the presence of the migrants in Alexanderplatz. Given his own personal history of forced displacement, followed by his experiences in communist East Germany, he occupies a distinctive position in German society, perceiving himself as both an insider and an outsider. While listening to the public debate, he even worries, 'Will they send him away for not being a local resident? He doesn't want to say who he is, or why he's here' (2017, p. 26), almost operating as an outsider in his home country. In this sense, the novel captures Erpenbeck's own sense of cultural dualism, given her upbringing in post-war East Berlin. In her 2020 memoir *Not a Novel*, she writes, 'I am not a refugee, but my past also took place in a different country, and it was a stroke of good fortune that the Federal Republic simply issued us new West German passports—something refugees today can only dream of' (Erpenbeck 2020, p. 179).

Focalised through Richard's denotive, phlegmatic calm and detachment, the novel documents the German public's reaction to the emergent humanitarian crisis while gesturing to the nation's longer, less discussed history of migration. In this context, *Go*, *Went*, *Gone* has faced some criticism for prioritising a Eurocentric outlook over the lived experience of the refugees (see Hermes 2016), thus positioning the novel within a longer history of German literature's partisan engagement with its colonial legacy. Such views are not without merit; the privileging of Richard's perspective—as a detached onlooker—brings to mind Joseph Slaughter's contention that the literary strategy of cultivating and prioritising 'a *noblesse oblige* of the powerful (rights holders) [...] 'ultimately reconfirms the liberal reader as the primary and privileged subject of human rights and the benefactor of humanitarianism' (Slaughter 2008, p. 104). The novel's lack of polyphonic form certainly supports this assessment and suggests Erpenbeck is merely contributing to the elision of the refugee voice, with Richard embodying the hegemonic structures responsible for such Western-centric bias.

Admittedly, Richard's intellectualisation of the brutal reality of forced migration (when confronted with the traumatic narrated events of his interlocutors) relies on his retreat back into the domain of classical myth, a form of knowledge production that enables Richard to retain an emotional distance from the traumatic actuality. As Yogita Goyal pertinently asks, 'If the refugee is a creation of law, what does it mean to render [them] myth, metaphor, or universal human?' (Goyal 2020, p. 250). Richard's renaming of the refugees as figures of Greek or mediaeval myth—a form of mnemonic congruous with his literary background—signals an intellectual repositioning of the refugee, remapping more global events as forms of European heritage and contributing to the elision of their true identities. Awad, a Ghanaian refugee, is christened Tristan, after Gottfried von Strassburg's 12th-century romantic hero, while Raschid is referred to as 'the thunderbolt-hurler' because his powerful nature and physique bears resemblance to the Greek Olympian Zeus (2017, p. 105). Aestheticizing the refugee within white frames of reference, simply because he 'finds it difficult to remember the strange names of Africans', he can only understand the plight of the refugee by recontextualising them within classical European paradigms, suggesting the limitations of Western empathy and epistemological thought (2017, p. 84). Richard's similar retreat into intellectual reclusivity, via a consideration of colonial history in order to understand the refugee protest, has also been criticised, notably his admission that he 'has no idea what German East Africa is called today' (2017, p. 37):

> 'Where exactly is Burkina Faso? [...] Some of his first-year students had been unable to recite even the first four lines of the *Odyssey* in Greek [...] He gets up and takes out his atlas. The capital of Ghana is Accra, the capital of Sierra Leone is Freetown, the capital of Niger Niamey. Had he ever known the names of these cities?'. (2017, p. 23)

It is possible to discern an authorial presence in this critique of Richard's geographic and colonial ignorance, compounded with his perception of the world through classic mythology and a European cultural imaginary through which to understand narratives of flight (Homer's *Odyssey* being his key point of reference, given its concentration on exile and displacement). Appropriately, the novel fluctuates between Richard's narration, the omniscient third-person narrator, and refugee voices that increasingly impinge on his

central account. Yet even in his ignorance, Richard endeavours to educate himself on the misconceptions and biases he has internalised through his reading of translated works of black literature on his bookshelves, developing an initial awareness of how Eurocentric cultural and pedagogic models have shaped his understanding of the inhumanity of colonial history (2017, p. 18).

Such a negative reading of the novel also neglects Erpenbeck's nuanced take on the politics of migration; her narratorial decision forces a consideration of the critical stakes behind such geopolitical decision making. Privileging the perspective of the cosmopolitan intellectual underlines that refugees *are* voiceless, pointing to the exclusionary mechanisms at work in deepening political invisibility and the precarious legal status of the refugees themselves. The figure of the refugee depends on the hospitality and goodwill of the European citizen in a moment of closing borders, their fate determined by the very nations and systems of neo-imperial control that led to the need for migratory movement in the first place. It is cultural empathy that is required, even though that process is hindered by Richard's nescience of the politics of migration. Narrative accounts of exile, displacement, or placelessness are rendered mute by forces of detention, which reduce refugees to a state of invisible immobility, their presence in European countries often corrupted to affirm national narratives celebrating cultural progress or new models of diversity.

The detention of refugees is a purposeful political act to prevent the debate from gaining traction or presence in the public sphere. With this in mind, the tactics used by the Alexanderplatz refugees to retain their visibility counter the political process of 'Othering' which attempts to delegitimise the demands of those at the margins of German society. The efforts by German authorities, 'defending their borders with articles of law', are in effect a ploy to 'assail these newcomers with their secret weapon called time', subjecting refugees to new forms of surveillance and detention, while ensuring they remain trapped by age-old states of abjection and rejection (2017, p. 81). Erpenbeck's narratorial decision thus gestures to Richard's inability (and by extension, the reader's) to bear witness, to identify with the pain, allowing the novel to turn away from banalizing experiences of migration; such flexibility also reflects an acute awareness of the limitations of representational depictions of trauma or suffering and the empathy they generate. Erpenbeck warns, 'Listening is an art—it is a risk—because those blind spots hide our own guilt and impotence', implicating the reader in this critique of superficial empathy—recognising the limitations of positionality—and calling for more active forms of substantive action (Erpenbeck 2018).

We can position Erpenbeck's novel alongside works such as *Spring* (2019) by Ali Smith or *Tell Me How It Ends* (2017) by Valeria Luiselli, which criticise bureaucratic forms of detention and deportation. As evident in those novels, Erpenbeck's refugees are also 'trapped between these now invisible fronts in an intra-European discussion that has nothing at all to do with [them] or the actual war [they are] trying to escape from' (2017, p. 68). The central problem with the Dublin Regulation, Richard decides, is that it 'doesn't concern itself with the question or whether or not these men are victims of war', resulting in 'asylum-fraud' and permitting 'European countries without a Mediterranean coastline to purchase the right not to have to listen to the stories of arriving refugees' (2017, p. 67). As Alison Mountz succinctly puts it, the 'immigrant-receiving states of the global North police borders and exacerbate differences between themselves and "others" who struggle to land on sovereign territory' (Mountz 2011, p. 255). Through a detailed discussion of EU asylum policy, which runs throughout the novel, Erpenbeck accentuates this conflictual tension between the traumatic personal memories of the refugees and governmental reports that selectively transmit the narratives that are recorded and those that elide lived experience in favour of a protective justification of archaic laws and regulations. Richard begins to operate as an ethnographer, as opposed to his previous acts of detached scholarly pursuit, conducting empirical research on the lives of those seeking asylum in his city. Despite his own liminal experiences in the German Democratic Republic, Richard's own methodical 'catalog of questions' in his initial interviews with the refugees not only reveal his own insular ignorance of global affairs, but also parodies the inadequacy of asylum interview

processes that fail to acknowledge the true trauma of forced migration: 'What vegetation is there in your country? Do people have pets? Did you learn a trade?' (2017, pp. 38, 47).

Indeed, *Go, Went, Gone* draws its power through the subtle acknowledgement of what is left out of the refugee story, only briefly alluded to by passing conversations with a Western stranger. Erpenbeck consequently unsettles rather than displaces the role or account of the intellectual—via the remedy of curative refugee storytelling—to articulate the role literature has to play in responding to such socio-cultural developments, with narrative hospitality bringing so arrestingly to light the absence of political hospitality in the contemporary moment. The rhythmic lives of the refugees reverberate through the narrative, despite their purposeful liminality at the margins of German society, ensuring their voices continue to be recorded and heard above the combative clamour of nationalist fervour. Richard's conversations with the refugees—although epistemically marginal and related in reported episodes—are a vital form of engagement and prevent the men from being reduced to empty statistics; as the writer Viet Thanh Nguyen insists, 'only though such acts of memory, imagination, and empathy can we grow our capacity to feel for others' (Nguyen 2018, p. 17).[2] Richard's dialogue, though initially naïve and uninformed in a state of strained colligation, represents a desire for the articulation of meaning across the seemingly impassable boundaries of race and cultural difference. In this sense, it is Richard's intention to intervene in the social world, no matter how flawed his attempts, that encapsulates the ethico-political tone of the novel, working across the fractured spaces of dissolution and displacement to create a stable sense of belonging for those not at home in the world.

By removing the figure of the refugee as a spectacle, enveloping them in quotidian lived experience, Erpenbeck counters the language of crisis that is often attached (and distorted) when discussing the politics of asylum, while retaining efforts to humanise the refugee and challenge established systems of border security and control. The narrative focalisation, then, rather than continuing the legacy of Eurocentric perspectives on modernity, prevents the reader from viewing the refugees as merely victims—defined by their discourse of trauma—but as fully rounded individuals, their lack of narrative centrality a reflection of their political voicelessness. Goyal claims it is precisely *because* 'so much of the representation of refugees in the media relies on spectacle, crisis, and catastrophe [that] the novel's concern with interiority and psychological depth, the cultivation of empathy, and the navigation of the relationship between an individual and the community can help counter such spectacularization' (Goyal 2020, p. 249). Further, by giving equal weight to the perspectives of German nationals, fearful of recent increases in migratory movements, Erpenbeck gestures to the cultural narrative of invasion that led to the rise of the AfP in Germany, and the reestablishment of the Fortress Europe mentality across the EU more broadly. Positioning such debates against Germany's recent fractured history of division, the novel exposes how national attachments and cultural imaginaries shape the political fates of refugees and continue a brutal legacy of racial exclusion.

Reframing the figure of the refugee thus involves a destabilisation of established native-immigrant binaries, a rethinking of the markers of citizenship, and an interrogation of the increasing frailty of secure national borders. Though Erpenbeck purposely accentuates the difficulties of political and artistic forms of representation, the novel indicates literature's capacity to engage in activist thought and generate a form of *productive authenticity*. The fusion of (auto)biographical testimony gathered from personal interviews, combined with authorial research into EU policy and a fictional reworking of the European literary figure (Erpenbeck, like her protagonist, was born in East Berlin), allows the novel to give voice to those narratives silenced by public or media xenophobia, or by the hostile institutional policies designed to curb refugee rights. That being said, Erpenbeck is far too subtle to suggest Richard's narrative arc from a detached onlooker to a politically engaged citizen is so simple or enlightened, nor is Richard forced into any supererogatory actions. The novel ends without any real change on a public scale, and Richard remains rather

ignorant of the neocolonial histories of European exploitation that have shaped these acts of forced displacement.

Superficial sentiments, Lyndsey Stonebridge reminds us, 'not even when ironically self-conscious about their own inadequacy, will never be enough' (Stonebridge 2020, p. 43). Yet in Richard's defence, he immediately recognises his inability to imagine the experiences of forced migration he encounters—this vertiginous, seemingly impassable distance—and moves beyond the traditional Western liberal protagonists, congratulating themselves on their acts of empathy and compassion. In persuading his friends and colleagues to house the refugees facing deportation—fellow East Germans who have experienced living beyond the wall—we see in Richard an outward movement from the reclusive, retired individual to an ethically engaged citizen. His concerted efforts to enact local socio-political change rely on Derridean forms of unconditional hospitality, whereby 'cities of refuge' have 'a duty to hospitality' where asylum is concerned (Derrida 2000, p. 4). Derrida's unconditional hospitality, of course, operates above the limits and laws of political hospitality, resisting the legislation pertaining to border enforcement and asylum. This microcosmic implementation of Merkel's controversial *Willkommenskultur* signals a turn to the communality required to tackle inward-looking, culturally bound perspectives on asylum policy, a direct authorial critique of Germany's growing right-wing political movement that stubbornly denies both empathy and obligation. Though the generosity of Richard and his friends is but a temporary reprieve in the fate of the refugees, who will likely be eventually deported or subject to further forms of detention, living suspended lives, their actions are more powerful for their limited scope. Such hospitality serves a deeper purpose, widening the scope of the German national identity to question why migrant memories and experiences are not permitted within the cultural imaginary. Could it be, Richard questions, that 'the people living here under untroubled circumstances and at so great a distance from the wars of others [have] been afflicted with a poverty of experience, a sort of emotional anemia?' (2017, p. 241). The novel's focalisation on the experiences and sites of trauma of those seeking asylum from *within* a German setting reinforces Erpenbeck's ethico-political intent. Rather than conceptualising some unrealistic sea-change in public attitudes towards immigration or a solution to structural impediments relating to European supranationalism, Erpenbeck focuses on small-scale changes at an individual level to confront the reader with palliative support solutions that render the experience of asylum more bearable.

## 3. Migratory Waves

Richard unconsciously shares a similar sense of dislocation with the refugees he so attentively monitors; his mother's expulsion from Silesia following the end of World War II, while Richard was just an infant, goes some way to explaining his fascination with the protests (as too does his wife's escape from Eastern Europe). Though Richard is perhaps not best placed to understand the traumas unfolding around him, Claire Messud acknowledges how his tumultuous childhood and East German connections ensure he is, at least, 'attuned to the potential of fleeing and needing assistance' (Messud 2017). His investigation into the lives of the refugees also becomes a personal journey of self-evaluation and commonality, contemplating the ways by which he himself has transitioned, following the fall of the Berlin Wall, from an East German citizen to a valued member of a reunified Germany:

> 'To investigate how one makes the transition from a full, readily comprehensible existence to the life of a refugee, which is open in all directions—drafty, as it were—he has to know what was at the beginning, what was in the middle, and what is now. At the border between a person's life and the other life lived by that same person, the transition has to be visible—a transition that, if you look closely enough, is nothing at all'. (2017, p. 39)

The very presence of the refugees also leads Richard to reinterpret the spaces of his daily life as a reenergised flaneur, excavating the deep roots of his adopted city. Following his conversations with Awad and Rashid, Richard reflects on the neo-colonial legacies and migratory flows that increasingly unsettle the fixity and assumed permanence of

national space: 'a border [. . .] can suddenly become visible, it can suddenly appear where a border never used to be: battles fought in recent years on the borders of Libya, or of Morocco or Niger, are now taking place in the middle of Berlin-Spandau' (2017, p. 209). Unsettling established paradigms and charting periods of historical transition, Erpenbeck's novel constitutes a form of cosmopolitan hospitality, accentuating how borders are continually politically and socially contested, bound up, and shaped by issues of identity or cultural belonging.

Michael Rothberg identifies that connecting events via a form of historical comparison is tempting; after all, 'we cannot *not* attempt to understand our local situation (whatever it is) without reference to global, historical developments in a variety of other national contexts', but doing so risks eroding or overlooking the political, social, or cultural specificities of those very developments (Rothberg 2019, p. 820). Richard's comparison of temporal scales relating to migration, linking the crossing points created by the Berlin Wall to the humanitarian crisis—'In 1990 he suddenly found himself a citizen of a different country, from one day to the next, though the view out the window remained the same'—does sporadically result in vague generalisations and allows him to overlook the true geopolitical drivers of asylum (2017, p. 81). However, this is clearly an authorial tactic to expose and accentuate his previous lack of awareness concerning this issue.[3] In conversation with Awad, a Ghanaian national previously living and working in Tripoli as a mechanic, Richard learns about the impact of Gaddafi's violence on ordinary citizens; in Awad's haunting account, his father is shot and his home destroyed in Libya's civil war, forced to join the multitude of refugees making unsafe passages across the Mediterranean: 'You have to drink. A few people died. There were sitting right there next to us, and then one would say, very quietly: my head, my head [. . .] and then the next moment he was dead. When people died we threw them in the water' (2017, p. 62). Rashid, a Nigerian metal worker, was also compelled to escape Libya on an overcrowded vessel carrying 800 people, barely surviving a shipwreck in which his children Amina and Ahmed both drowned. Confronted with the brutal histories behind forced migration, Richard bemoans how he is supposed to understand the geopolitical specificities that inform each case, which are often elided in media reports of immigration, 'how many coverings must be torn away before he's finally able to truly grasp things, to understand them to the bone?—Is a human lifetime long enough? His lifetime or anyone else's? (2017, p. 142).

The third-person omniscient narrator that underpins Richard's narration subtly alludes to shared experiences of migration that inform German history and the potential for entangled solidarities that may emerge from such understanding, but importantly stops short of forcing an inaccurate comparison between the plight of African refugees and those citizens still coming to terms with German reunification. Rather, the palimpsestic nature of Berlin over the last four centuries, when considered against the uneasy colligation of Richard's personal memories, repositions the act of migration as something firmly rooted in German cultural history and contradicts the oft-quoted political misconception that Germany is 'not an immigration country'; '*Kolonialwaren* [Colonial Goods] and WWII bullet holes might adorn the very same storefront' (2017, p. 36). For Geddes and Scholten, the reasons why Germany does not view itself as an 'immigration country' are 'deeply rooted in the very specific history of Germany as a divided nation for a long period after the Second World War', strengthening Erpenbeck's concentration and comparison of the country's historical periods (Geddes and Scholten 2016, p. 99). Huguenot refugees fleeing France, Richard recalls while walking through Oranienplatz, were 'the original settlers' of the streets; yet locals are shocked at the transformation of the Alexanderplatz into a space for refugee protest, 'Now the square looks like a construction site: a landscape of tents, wooden shacks, and tarps' (2017, p. 32). Though Richard remains somewhat ignorant of the deeper cultural legacy of German colonial exploits, despite his own childhood memories of forced migration, the novel gestures to the power of *multidirectional* memory via its consideration of Germany's twentieth-century historical tragedies. Coined by Michael Rothberg, multidirectional memory produces a 'productive, intercultural dynamic', creating

the potential for 'new forms of solidarity and new visions of justice', an intent that is deeply embedded in Erpenbeck's literary endeavour (Rothberg 2009, p. 5).

We can therefore position Erpenbeck within a wider movement in the European literary canon, with writers acknowledging the longitudinal effects of migratory waves and the complicity of nation-states in deepening radical inequalities of access: 'This movement of people across the continents has already been going on for thousands of years, and never once has this movement halted' (2017, p. 143). Richard's 'shift in his perspective and sense of scale' forces him to abandon his chthonic fascination with European heritage and mythology and, instead, ethically invest in the shared global concern of forced migration and asylum (2017, p. 55). Returning from the refugee hostel, he perceives his old life of routine and domesticity as that of a stranger 'strolling through a museum, as if he himself no longer belonged to it [. . .] room after room, suddenly appeared to him utterly foreign, utterly unknown, as if from a far-off galaxy. His tour ended in the kitchen' (2017, p. 91). In a 2018 public keynote at the Puterbaugh Festival, Erpenbeck spoke of the need for Western citizens to acknowledge these 'blind spots', to 'step back in order to see an entire historical tapestry extending far beyond your own lifetime' and consider the longue durée of migration (Erpenbeck 2018). Recounting a visit to Princeton University for a reading, she recalls:

> 'I met a professor who proudly mentioned that she had urged one of Angela Merkel's advisers to accept Syrian refugees. But her advice was not that Germany should accept the refugees instead of rejecting them; rather, it was that if Germany had to accept refugees at all, then at least it should accept Syrian refugees instead of others who were less well educated and sophisticated. And they had to act fast before other countries beat them to the punch. She told Merkel's adviser that Syrians were elite as far as refugees were concerned; they'd be the easiest to integrate and the quickest to pay back into the system. That way Germany wouldn't have to accept those poor suckers, you know, the ones from the "shitholes"—like Niger, for example'. (Erpenbeck 2018)

Though the refugees hail from African countries (Erpenbeck finished the novel before the Syrian civil war led to further issues in the Mediterranean), she acknowledges how the events of the narrative anticipate Europe's collective failure to address the mounting tensions surrounding immigration.

Speaking simultaneously to two distinct historical and political moments, offering a commitment to the occupations of multiple positions in forging a sense of commonality and ethical engagement, the term thus aligns with the transglossic notion of *deep simultaneity*, forging connections between the parallel experiences of forced migration throughout the novel (Shaw and Upstone 2021, p. 581). Within this hauntological context, a space emerges that enables the protagonist to recognise his own personal traumas of displacement: 'Becoming foreign. To yourself and others. So that's what a transition looks like' (2017, p. 63). Richard forces himself to confront the intimate realities of the Other, even though his depth of understanding—and attempts to move beyond existing frames of national reference—will always be confined by the historical paradigms of which he is a part. The spectres of the past emerge not as a barrier to the present, but as a means of recognising the legacies that unite global citizens across the entrenched barriers of race and nationality. Germany's colonial and neocolonial legacy thus functions as a spectre that returns, imbuing the novel with a hauntological edge through the spatial remapping of excavated cultural histories.

The novel's title itself holds clues to Erpenbeck's ethical intent. In language acquisition lessons designed to tackle social integration and welfare dependency, 'the only thing the Berlin Senate will still pay for in the case of the men who now aren't supposed to be here at all', the refugees learn the conjugation for the verb of motion 'go' (2017, p. 272). While the present tense implies momentum and progress, the subsequent past tense 'went' (*ging*) and past perfect 'gone' (*gegangen*) not only allude to this multidirectional memory, which impacts the narrative, but also to the absence of a certain future for the refugees. The title's grammatical oppositions between past and present tenses insinuate how the refugees'

immobility continues to be impacted by personal traumas and ongoing detention for which there is no end in sight, as indicated by the absence of the present perfect continuous 'going'. The verb of mobility, to go, is thus reduced to a verb of stasis and uncertainty. Suspended in bureaucratic abeyance, the refugees lead limited lives 'in which an empty present is occupied by a memory that one cannot endure, in which the future refuses to show itself' (2017, p. 277). Indeed, Erpenbeck breaks from the novel's rather stoic and stable narrative rhythm to play with typography in reinforcing this message, repeating the question, 'Where can a person go when he doesn't know where to go?' (Erpenbeck 2017, pp. 266–67). The refrain, positioned in the centre of two blank consecutive pages, forces the reader to contemplate the emptiness and silence of a response, and the failure of the German state or the EU more broadly to navigate a way through these developments and find a viable solution.

The transglossic qualities of Erpenbeck's authorial responsibility are also evident in her acknowledgments, which contain information on how readers can donate to refugee charities, strengthening the positioning of the novel as a critical report into the state of German asylum policy. As with Ali Smith's *Seasonal Quartet*, Viet-Thanh Nguyen's *The Committed* (2021), or David Herd's *Refugee Tales* collections, we see in Erpenbeck's body of work an effort to be *explicitly*, rather than implicitly, political; writers are employing the novel as a means of giving legitimacy to the refugee voice and serving as activists dedicated to the protection of human rights. As Nguyen argues, 'literature does not change the world until people get out of their chairs, go out into the world, and do something to transform the conditions of which the literature speaks' (Nguyen 2018). Richard comes to understand that his cultural hospitality is ineffective unless it is rooted in the communal. Grounded in her personal interviews with refugees in Germany, and a detailed understanding of European Union law, *Go*, *Went*, *Gone* thus captures what I term Erpenbeck's *compassionate politics*. Following Hunt (2007) and Slaughter (2008) in recognising the central role of empathy in the act of reading, we can perceive in Erpenbeck's narrative logic a distinct humanist vision aligned with an acute understanding of the failures of rights legislation. Looking towards the future, Erpenbeck's narrative forces a consideration of how literature may respond to or adapt to the figure of the refugee in an era of revanchist nationalism and economic inequalities. For Giorgio Agamben,

> 'given the by now unstoppable decline of the nation-state and the general corrosion of traditional political-juridical categories, the refugee is perhaps the only thinkable figure for the people of our time and the only category in which one may see today [...] the forms of limits of a coming political community'. (Agamben 2008, p. 90)

And yet, the refugee is treated with such caution and fear precisely because it undermines the legitimacy of the nation-state and challenges the parameters of established cultural imaginaries.

## 4. Bureaucratic Geometry

Erpenbeck's most acute critique is saved for Germany's legal system—and, by extension, the EU's Common Asylum System—which protects itself from evolving global developments by employing empty rhetoric even as it purportedly attempts to solve the humanitarian crisis. While researching German asylum policies on the internet, Richard discovers the word *Fiktionsbescheinigung*, a piece of paper certifying one can stay in the country without a residence permit, ruminating on how this certificate of *fiction* amounts to very little, 'merely a confirmation that this person existed who had not yet been granted the right to call himself a *refugee*' (2017, p. 82). Accompanying his friend Ithemba to a lawyer's office to determine whether he is entitled to the exceptional right to remain, Richard realises that the trials and tribulations of German asylum paperwork and procedure are just as perilous as the Mediterranean crossings the refugees have endured; he is astounded to learn 'a leave to remain is just a *temporary suspension of deportation*', trapping the refugees in a Kafkaesque nightmare (2017, p. 249). In their subsequent debate on the finer details

of the Residence Act, the lawyer reminds Richard of the teachings of Roman historian Tacitus, whose renowned ethnographic work *Germania* accentuated the need for ethical hospitality: 'It is accounted a sin to turn any man away from your door [...] It makes no difference that they come uninvited; they are welcomed just as warmly. No distinction is ever made between acquaintance and stranger as far as the right to hospitality is concerned' (2017, p. 251). Such openness and conviviality have been eroded by written laws that 'come to replace common sense' and are no longer 'anchored in the emotional lives of the people [...] [being] formulated with such a high level of precision and abstraction that all basis in human emotion has become superfluous' (2017, p. 250). Tacitus's legacy, the lawyer concludes, is that 'we're left with section 23, paragraph 1 of the Residence Act' (2017, p. 251).

Such perverse and unintelligible forms of 'bureaucratic geometry' have deep roots; the colonised, Richard ruminates, were similarly 'smothered in bureaucracy' (2017, p. 49). However, heightened public opposition to immigration more generally undoubtedly compounds the problem, with the humanitarian crisis provoking anger, not empathy, from his fellow countrymen, 'Someone on the internet calling himself *DontCare* writes: The only ones I feel sorry for are the coast guard workers! Why should they have to keep going out there to drag bodies out of the water?' (2017, pp. 166–67). The spread of xenophobic discourses that pathologise forced migration, positioning refugees as the source of their own misfortune, typify the political mood, resulting in arbitrary political categorisations, public labelling, and misconceptions surrounding the act of asylum:

> The Africans have to solve their problems in Africa, Richard's heard people saying many times in recent weeks [...] For a moment, Richard imagines what a to-do list would look like for the men he's gotten to know over the past few months.
>
> His own to-do list would look something like this:
>
> Schedule repairman for dishwasher
>
> Urologist appointment
>
> Meter reading
>
> [...]
>
> And for Rashid the list would read:
>
> Broker a reconciliation between Christians and Muslims in Nigeria
>
> Persuade Boko Haram to lay down their arms (2017, pp. 203–4).

The untranslatability of migration ensures there is a clear empathy deficit to the humanitarian crisis while, at a geopolitical level, Eurocentric systems of inclusion and exclusion continue to dictate affairs, with potential cross-cultural connections offset by bleak cosmopolitical realities. We witness in Erpenbeck's novel a pronounced resistance against what David Herd has termed 'a politics of expulsion' that increasingly utilises detention as a means of exploiting the state of 'geopolitical non-personhood' (Herd 2023, p. 3). Despite Richard's best efforts, letters from the Foreign Office confirm the vast majority of the refugees are denied asylum due to the brute reality of the Dublin II regulations, returning them to the first country they entered in the EU; only 12 exceptions are made out of 476 cases. As a result, Richard is provoked into indignant opposition to asylum policy, as opposed to his previous state of superficial compassion and empathy for their plight.

In their analysis of the contemporary political landscape, Krastev and Leonard (2024) identify Germany as the only country in Europe where immigration is listed as the biggest threat to the future in public polls, ahead of other polycrises such as climate change, the war in Ukraine, global economic recession, or the impact of COVID-19. Recent polling suggests the majority of Germans continue to prefer fewer refugees to be admitted into the country (see Kinkartz 2023).[4] Through her detailed engagement with asylum policy, Erpenbeck indicates how there is a clear and obvious failure by supranational institutions such as the EU to share and delegate responsibility in a fair and measured manner when dealing with matters of asylum. Neither Schengen nor the Dublin Regulation are suggested to be

feasible solutions in addressing new developments in global migration. Erpenbeck points towards the need for a long-term, holistic response to the drivers of migration rather than a disjointed and knee-jerk reaction to border control and asylum policy. For Crawley et al. (2018), despite Merkel's policy of *Willkommenskultur*, the subsequent failure of Germany to provide hospitality to the refugees is a reflection of the deep-seated financial issues facing the European Union and the long-standing multilateral failure to provide humanitarian aid. *Go*, *Went*, *Gone* therefore gestures to a wider EU crisis concerning public backlash to the erosion of the national sovereignty of member states, which in turn fuels more repressive forms of asylum policy and garners support for illiberal political forces.

A subsequent robbery at Richard's house in the closing stages of the novel, intimated to have been the work of the only refugee who possessed prior knowledge of Richard's trip to a conference in Frankfurt, does little to muddy the water of Erpenbeck's compassionate politics. Richard suspects that Osarobo may have broken into his house when he was away, but he shies away from raising the issue. His friend Anne explains to him why he should 'make a scene': 'you have to take him seriously. If you make excuses for his betrayal, then you're basically just putting on airs, playing the morally superior European' (2017, p. 257). The phlegmatic detachment and emotional restraint of Richard's narrative perspective prevent the novel from adopting an overly didactic or moralistic tone; tellingly, the concluding scene of the novel culminates, in a circular return, to Richard's personal meditation on his own life and what he has lost, rather than supplying a grandiose statement indicative of Richard's ethical paradigm shift as a white savoir or giving the final word to his disenfranchised interlocutors. The novel's lack of closure is reflective of the absence of a central authority in dealing with forced migration and the radical uncertainty that emerges in its wake.

Empathy has limits, Erpenbeck reminds us, yet such limitations should not direct citizens to apathy or insularity. As Nussbaum argues, the novel as a cultural medium 'generally constructs empathy and compassion in ways highly relevant to citizenship'— qualities that become more urgent in light of global migratory developments (Nussbaum 1995, p. 10). Narrative empathy has the potential to move us to moral action if it succeeds in transforming our sense of what kind of moral agents we are and want to be.[5] It may even produce a collective political impact if it engenders a transformation of the cultural narrative imagination. By forcing a confrontation with latent European histories of migration, Erpenbeck forges a more inclusive communal memory that changes the parameters of the German cultural imaginary, redefining who belongs in the nation. Erpenbeck thus invites the reader into her ethical project, indicating the empathetic power of the novel in bestowing a model of narrative hospitality that reframes the borders of the mind.

**Funding:** This research received no external funding.

**Data Availability Statement:** No new data were created or analyzed in this study. Data sharing is not applicable to this article.

**Conflicts of Interest:** The author declares no conflict of interest.

## Notes

[1]  See Bartels (2015). Fazila Bhimji's (2016) research on refugee activism in Berlin also speaks to the very same protests that influenced Erpenbeck's novel.

[2]  Martha Nussbaum (1997) has accentuated the importance of immigration in generating empathetic identification with otherness.

[3]  Erpenbeck has spoken of how the humanitarian crisis resulted in a confrontation with her own profound sense of displacement following the end of the GDR: 'I was in my early twenties when the Berlin Wall fell and the country where I had grown up disappeared in the course of just a few weeks' (Erpenbeck 2018).

[4]  78% of respondents claimed that the integration of refugees into Germany was not working well (Kinkartz 2023).

[5]  Though empathy and compassion appear as rather ineffectual tools in resolving global migration, Dominick LaCapra (2001) and Suzanne Keen (2007) have indicated the ways by which writing, as a form of social activism, can lead to political change.

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
