# Peer review of "Reframing the Refugee: Jenny Erpenbeck’s Compassionate Politics"

_humanities, doi:10.3390/h13020047_

Round 1

Reviewer 1 Report

Comments and Suggestions for Authors

The article is a well-written and critically supported analysis of Jenny Erpenbeck’s novel Go, Went, Gone within the theoretical framework of migration and refugee studies. The topic is original in the attempt to filter through a recent literary work contemporary political and ethical issues and is consistent in showing how the author reflects through her novel her compassionate politics. The language is fluent and concepts are clearly expressed and this makes the text easy to read.

-The conclusions of the article are slightly hasty (especially the concept of ‘narrative hospitality’) but still they address the questions and issues posed at the beginning. However, it might be useful to elaborate on Derridda’s concept of unconditional hospitality that is only briefly mentioned in line 321.

-Also Bhabha’s endorsement of the novel (quoted in lines 64-67) could be elaborated on in relation to his observations on how the novel tackles the theme of migration.

-The author should also make it clear in lines 86-88 that the narrator of the novel is not the protagonist but a third-person omniscient narrator focalized on Richard’s perspective. This is hinted at later (line 396), but it should be specified and better explained here where the first quote from the novel is inserted.

-The very long sentence in lines 107-112 should be revised as it makes the concept expressed quite confused.

- the reference to “some criticism” the novel has faced (line 159) seems incomplete and could be more widely expanded: it is not clear whether or not the author supports some of the issues presented by criticism.

Some minor typos and syntactic oversights are:

Line 43: a comma is missing in the novel title

Line 72: titles of novel are graphically bigger. The same is in line 232 (beginning of quotation)

Line 139: “than” is missing

Line 401: eliminate “the”

Line 468: reference in brackets should go after the quotation

Line 487: Seasonal Quartet should be in italics

Line 518: “how this” instead of “this how”

Author Response

I have corrected the minor typographical/grammatical errors identified by both reviewers.

I have also added more material when requested by Reviewer 1:
Lines 67-69 - expanding on Bhabha and his commentary of the novel.
Lines 326-328 - unpacking Derridean 'unconditional hospitality'

Reviewer 2 Report

Comments and Suggestions for Authors

I believe the present essay, on Jenny Erpenbeck’s ‘Go, Went, Gone’ (2015; English trans. 2017) is publishable as long as the author engages in some revision. My comments on the essay’s first few pages raise minor questions about some of the author’s phrasings. Subsequent comments, having to do with the latter half of the essay, concern ideas I found unconvincing. I believe it is important that the author address both groups of comments, and I hope they take seriously my scepticism about the essay’s conclusions – particularly the idea that reading Erpenbeck somehow leads to political activism. 

Line 24 refers to the Schengen Agreement and the Dublin Regulation, which are now over 35 years old. Citing newer regulations governing Europe’s borders and keeping fortress Europe intact would be more convincing here. (For example, the Dublin III Regulation [2013] is apparently scheduled to be replaced by the Asylum and Migration Management Regulation (AMMR) in 2024).

Line 31 doesn’t provide relevant dates for context, which would help. The word ‘Willkommenskultur’ entered the discourse (at least in prominent use) in 2014/2015, and the first major uses, in the sense meant here, were in 2015. If Erpenbeck’s novel was already published in 2015, then she was likely writing in response to a general mood or crisis, but her thinking antedated the widespread use of the term and, significantly, Merkel’s public and highly controversial responses, which were at the centre of public debate from Sept. 2015 to mid-2016. 

Line 47: I don’t believe the novel should be described as semi-autobiographical. The fact that Erpenbeck conducted interviews from 2012 to 2014 means that she researched her novel, not that there was a significant autobiographical component. Barring other statements, I would dispute that characterization. 

Line 86: the phrase ‘Erpenbeck’s novel is marked by interiority’ – what novel isn’t? I’m not sure what the point here is. Nearly every novel begins with a person who is thinking about something, no? 

Line 98: the phrase ‘in Alexanderplatz’ is incorrect, because of the preposition. The text should read: ‘… at Berlin’s famous Alexanderplatz’ or ‘on the Alexanderplatz in Berlin.’ By the way, the Alexanderplatz, as the author surely knows, was in the former East and would have been highlighted by Erpenbeck for that reason. 

Line 104: tell the reader why Richard is arriving at the Oranienplatz (in another part of town, famously what was traditionally known for being the Turkish-German part of town, in former West Berlin), when the protest was at the Alexanderplatz. Yes, it’s accurate to the novel – a tent city is there – but that part of the storyline hasn’t been conveyed to the reader.  

Line 106: it’s worth mentioning that this has historical referents, referring specifically to when more than 360 refugees/ asylum seekers died near Lampedusa on 3 October 2013. 

Line 202-203: how does the translator deal with the phrase ‘N---rliteratur’? That’s an uncomfortable phrase in German, even if it doesn’t translate directly to the US’s N-word. 

Line 265: I don’t understand what is meant by ‘rhythmic lives.’ That seems like a debatable turn of phrase here. 

Line 270: I suspect that Viet Thanh Nguyen would not be enthusiastic about Erpenbeck’s decision to mediate the story through the eyes of a privileged (retired, white) ethnic German. Nguyen is cited here, but not really engaged with, and I gather that his overall view would be similar to that of Slaughter, ultimately objecting to how the liberal reader is the true subject of human rights, etc. (see lines 164-65). I don’t feel that these objections are adequately accounted for in the present approbation of Erpenbeck’s novel. 

Line 368: In my opinion, the reference to Lukacs is misplaced. As exciting as ‘Theory of the Novel’ is, transcendental homelessness has to do with an inheritance from Fichte and Marx, among others, and has, in my view, little to do with the present reading. I would suggest setting that aside. 

Line 401: typo ‘colligation with Richard’s’ (delete ‘the’). 

Line 417 is where the argument gets into trouble for me, particularly the presumption that reading literature leads to action. There’s no evidence for this. It’s a fantasy, I think, that reading the novel, simply because Erpenbeck tells people where to donate money, is the same as making activists out of readers and getting people ‘out of their chairs’ (see line 492). That’s thoroughly and entirely unconvincing. Moreover, what happens if people get out of their chairs, and we don’t agree with them? Along these lines: the author, regrettably, cites Giorgio Agamben affirmatively (see lines 504-507), but Agamben’s politics have since become reprehensible (see, for example, the 2022 article ‘What Happened to Giorgio Agamben?’). I’m sure all his years of reading got him out of his chair, but to what end? The author of the present essay assumes that all readers (of this essay, of Erpenbeck) are on the same page, in agreement about the refugee crisis, and that they will read Erpenbeck’s book and take to the streets. There’s nothing to back this up. It's terrific that Erpenbeck tries to be explicitly rather than implicitly political, by telling people where to donate, but good luck. If her mission is to mitigate the refugee crisis in Germany, writing a touching novel about a retired classics professor who eats toast and honey for breakfast isn’t that effective, at least not in comparison with working for the IRC, GRC, etc., something that most literature students, even the best ones, don't bother with. 

Along similar lines, the author writes (on line 610) that, ‘Narrative empathy has the potential to move us to moral action’ – I’m not seeing it. I think that’s a bit of a self-congratulatory fantasy of literature students, that reading is the same as politics. To choose one example: I’m sure that, at the Black Lives Matter protests in the US in 2020 a negligible percentage of participants – a percentage that probably rounds down to zero – were English literature students motivated to participate because they had read Toni Morrison. I love Toni Morrison, but I don’t mistake being moved by a novel (especially a highly erudite and challenging literary one), for activism. People reading the writings of, say, Fanon or the ‘Autobiography of Malcolm X’ is perhaps another story, but fortunately no one needs a graduate seminar in literature to be inspired by those works; people with nearly any degree of education can find them at the library. 

Finally, I’ll add that I don’t go along with the demonization of Germany (lines 572-577) as uniquely poor handlers of the crisis. Germany has done as well or better with its immigration anxiety than have many of its neighbours. The Sweden Democrats and the Brothers of Italy have performed better in their respective elections than Germany’s AfD. At least until this coming summer (we will see what happens, I suppose), Germany is doing better than many of its neighbours at holding its far right at bay. 

This is a good essay on Erpenbeck, but I have problems with its concluding linkage of the literary and the political. I hope the author sees fit to revise in light of these questions. 

Author Response

Reviewer 2:
Line 23-28 - outlining the development of the Dublin Convention and its subsequent modifications into new EU law.
Line 31 - Explaining how the novel was written before Merkel's Willkommenskultur (thus anticipating rather than responding to events).
Line 43-44; 49 - Correcting the statement that the novel is "semi-autobiographical" (it is rather informed by Ernpenbeck's own ethnographic work).